

# The accuracy of Random Forest performance can be improved by conducting a feature selection with a balancing strategy

Maria Irmina Prasetiyowati[1], Nur Ulfa Maulidevi[2] and Kridanto Surendro[2]

[1] Doctoral Program of Electrical Engineering and Informatics, School of Electrical Engineering and Informatics, Institut Teknologi Bandung, Bandung, Jawa Barat, Indonesia
[2] Department of Electrical Engineering and Informatics, School of Electrical Engineering and Informatics, Institut Teknologi Bandung, Bandung, Jawa Barat, Indonesia

## ABSTRACT

One of the significant purposes of building a model is to increase its accuracy within a shorter timeframe through the feature selection process. It is carried out by determining the importance of available features in a dataset using Information Gain (IG). The process is used to calculate the amounts of information contained in features with high values selected to accelerate the performance of an algorithm. In selecting informative features, a threshold value (cut-off) is used by the Information Gain (IG). Therefore, this research aims to determine the time and accuracy-performance needed to improve feature selection by integrating IG, the Fast Fourier Transform (FFT), and Synthetic Minor Oversampling Technique (SMOTE) methods. The feature selection model is then applied to the Random Forest, a tree-based machine learning algorithm with random feature selection. A total of eight datasets consisting of three balanced and five imbalanced datasets were used to conduct this research. Furthermore, the SMOTE found in the imbalance dataset was used to balance the data. The result showed that the feature selection using Information Gain, FFT, and SMOTE improved the performance accuracy of Random Forest.

## INTRODUCTION

Higher accuracy and quicker processing time must be considered in order to build a model. Unfortunately, those two are contradictory because any effort to increase the accuracy of one affects the processing speed and accuracy of the other. Therefore, this study determined the accuracy-performance and the required time to improve feature selection by integrating Information Gain (IG), the Fast Fourier Transform (FFT), and Synthetic Minority Oversampling Technique (SMOTE).

Random Forest is a classification algorithm based on the random selection of trees (*Gounaridis & Koukoulas, 2016*; *Prasetiyowati, Maulidevi & Surendro, 2020a*; *Prasetiyowati, Maulidevi & Surendro, 2021*), thereby making it uninformative as a tool used to build the decision tree (*Breiman, 2001*; *Prasetiyowati, Maulidevi & Surendro, 2021*; *Scornet, Biau*

Corresponding author
Maria Irmina Prasetiyowati,
33218014@std.stei.itb.ac.id

& Vert, 2015). However, this process allows the selected feature to be uninformative. Therefore, improving the feature selection process is necessary to make it informative with a faster execution time. Several studies have proposed the feature selection process for Random Forest (Adnan, 2014; Prasetiyowati, Maulidevi & Surendro, 2021; Sun et al., 2020; Ye et al., 2013; Zhang & Suganthan, 2014), including the use of IG with a threshold based on the standard deviation value (Prasetiyowati, Maulidevi & Surendro, 2021). Zhang & Suganthan (2014) proposed a new method in Random Forest by increasing tree diversity by combining a different rotation space at the root node. Ye et al. (2013) researched feature selection for Random Forests using the stratified sampling method, and the results showed the enhanced performance of Random Forest.

The number of features in a dataset varies from few to more than 100 features. However, not all features are informative, irrelevant, and redundant (Lin, Hung & Wei, 2018); therefore this affects the performance and accuracy (Chandrashekar & Sahin, 2014). One of the methods used to solve this problem is the Information Gain (IG), an essential technique for weighting the maximum entropy value (Chandrashekar & Sahin, 2014; Elmaizi et al., 2019; Jadhav, He & Jenkins, 2018; Nguyen, Shirai & Velcin, 2015; Odhiambo Omuya, Onyango Okeyo & Waema Kimwele, 2021; Singer, Anuar & Ben-Gal, 2020). According to preliminary studies, IG reduced the entropy value before and after the separation process and was used to determine the possibility of using or discarding an attribute. For instance, those equal to or greater than a predetermined threshold value of 0.05 are selected in the algorithm classification process (Demšar & Demsar, 2006; Yang et al., 2020). Sun et al. (2020) used the calculation of the threshold value of 0.5 as a determination of the occurrence of landslides. Landslides occur if the predicted value is greater than 0.5. Several other studies use the calculation of the frequency of each feature to determine the threshold value as a subset of the final features (Tsai & Sung, 2020). However, some also use the standard deviation to determine the threshold (Prasetiyowati, Maulidevi & Surendro, 2021; Sindhu & Radha, 2020).

Furthermore, the preliminary study shows that the standard deviation method, which aims to determine the threshold value did not calculate the class balance in the dataset. Therefore, this led to the development of several techniques to overcome this process. One of which is using the Synthetic Minority Oversampling Technique, also known as SMOTE (Chawla et al., 2002; Feng et al., 2021). SMOTE (Juez-Gil et al., 2021; Li et al., 2021; Mishra & Singh, 2021; Zhu, Lin & Liu, 2017), an excellent oversampling technique that reduces the risk (Chawla et al., 2002). However, SMOTE tends to cause problems when applied to unbalanced multiclass data, with generalization acting as a more severe problem and one of the minority classes to the majority (Zhu, Lin & Liu, 2017). The SMOTE stages are as follows (Feng et al., 2021):

1. Prepares the number of synthetic minority class instances
2. Selects a minority class instance randomly
3. Uses the K-Nearest Neighbor (KNN) algorithm to get associated neighbors from the selected instance
4. Combines minority and selected neighboring class instances to generate new synthesis by random interpolation.

Steps 2 and 4 are repeated until the desired amount is obtained.

This study followed previous studies (*Prasetiyowati, Maulidevi & Surendro, 2021*; *Prasetiyowati, Maulidevi & Surendro, 2020a*; *Prasetiyowati, Maulidevi & Surendro, 2020b*). The researchers began this study by using the Correlation-based Feature Selection (CBF) for feature selection. This study resulted in the time required by the Random Forest (RF) that was less than the study without performing the feature selection. However, the accuracy was poor (*Prasetiyowati, Maulidevi & Surendro, 2020a*). In the second study, the researchers continued to use the CBF. However, the dataset used in the study was the dataset that had been transformed using the Fast Fourier Transform (FFT) and reverted by using the IFFT. This study resulted in a better accuracy value than previous studies. The average accuracy value for the dataset that had been transformed increased by 0.03 to 0.08% compared to the original dataset (*Prasetiyowati, Maulidevi & Surendro, 2020b*). Even though the required time in this second study was shorter than that of the RF without feature selection, the total time did not include the time needed for transforming the dataset. The third study used the gain information with the threshold based on the standard deviation, fixing the required time and accuracy value (*Prasetiyowati, Maulidevi & Surendro, 2021*). This third study resulted in better accuracy than the previous studies, and the required time was also better. Nonetheless, the accuracy obtained from the study could not be superior to that of RF without feature selection. This study was only superior in the aspect of required time. The need for the increased accuracy value stimulated the researchers to implement the FFT to the feature. Based on the previous studies, FFT could improve i the accuracy value (*Prasetiyowati, Maulidevi & Surendro, 2020b*). In addition, this study also proposes integrating Information Gain, Fast Fourier Transform (FFT), and Synthetic Minority Oversampling Technique (SMOTE) algorithms to improve the accuracy of Random Forest performance. The FFT is used to transform feature values into complex numbers consisting of imaginary and real numbers, while the SMOTE is used for class imbalance problems and increasing accuracy values. Features with real values are taken, and the median value is calculated to determine the threshold. The stages or the roadmap of this study can be seen in Fig. 1. We also use the confusion matrix to analyze accuracy (*Sun et al., 2021*; *Zhou et al., 2021*).

This study is organized as follows: 'Materials & Methods' and 'Results' describe the related research and proposed method. Meanwhile, the results and comparisons with other methods and analyses are described in section 'Discussion'. Finally, the research conclusion is discussed in 'Conclusions and Future Work'.

## MATERIALS & METHODS

This study proposed a feature selection method using the median of Information Gain (IG), transformed with Fast Fourier Transform (FFT) to obtain real and imaginary values. However, the real values were taken to calculate the median of the IG, which are used to determine the threshold (cut off) subsequent processes. The equation used to calculate the IG value is shown in Eq. (1).

$$\text{gain}(y, A) = entropy(y) - \sum\nolimits_{C \varepsilon nilai(A)} \frac{Yc}{y} entropy(yc). \tag{1}$$

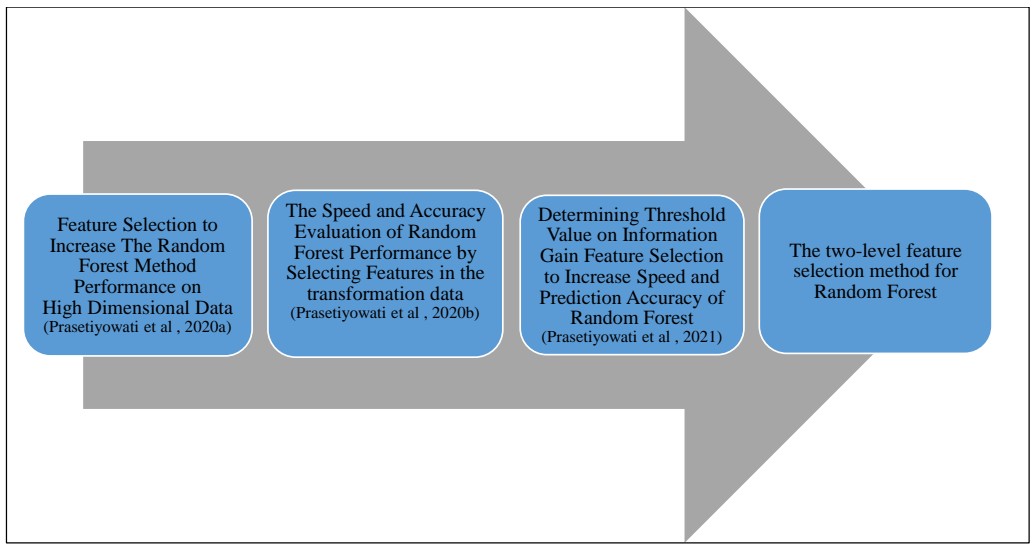

**Figure 1** **Roadmap of research** (*Prasetiyowati, Maulidevi & Surendro, 2020a*; *Prasetiyowati, Maulidevi & Surendro, 2020b*; *Prasetiyowati, Maulidevi & Surendro, 2021*)

The value c is an attribute, and Yc is a subset of y. The rule of Eq. (1) is the total entropy y, obtained after splitting the data based on feature X.

In the next step, the Information Gain value is transformed using FFT as in Eqs. (2) and (3).

$$X[k] = \sum_{n=0}^{N-1} X[n] W_N^{kn}, \quad k = 0, 1, \ldots N - 1 \tag{2}$$

where $W_N^{kn}$ referred to as the twiddle factor, has a value of $e^{-j\frac{2xkn}{N}}$, hence

$$X[k] = \sum_{k=0}^{N-1} X[n].e^{-j\frac{2xkn}{N}}, \quad k = 0, 1, \ldots N - 1. \tag{3}$$

The IG transformed by FFT is a complex number consisting of imaginary and real values. This study used the real value of the transformation results to calculate the median, the middle value that divides data into two (half). The median equation is seen in Eq. (4).

$$Median = data\frac{n+1}{2} \tag{4}$$

where $n$ is the number of data determined from the real value of the IG. After obtaining the median value, the next step is to cut off a threshold based on the median value. However, when the IG value is greater than or equal to (>=) the median, it is included as the selected feature.

Furthermore, this study also proposes using SMOTE for multiclass datasets with only two classes, namely the minority and majority. The SMOTE only synthesizes the minor data to balance with the major, intead of the minor. Furthermore, this study proposes

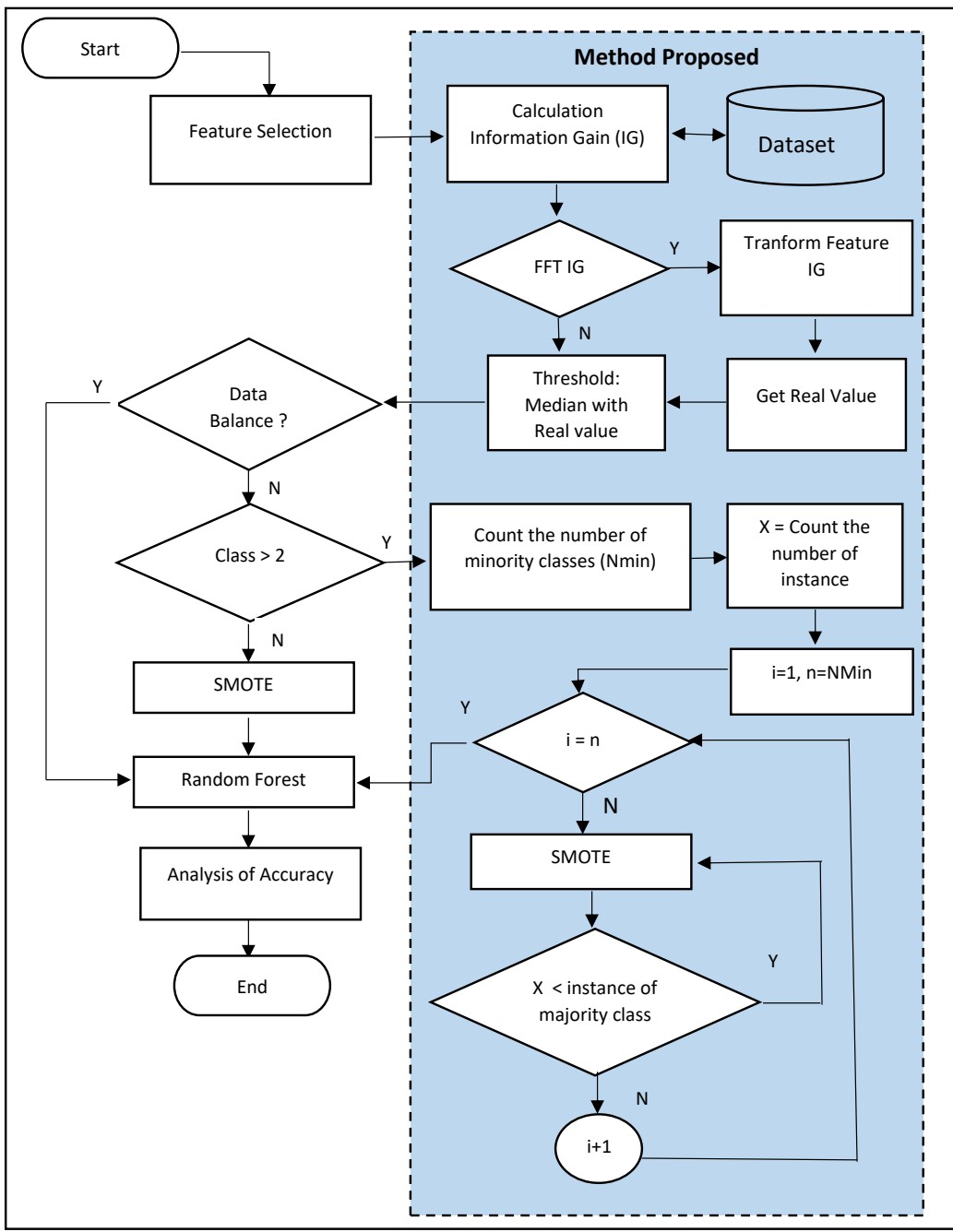

**Figure 2    Flowchart of the proposed method.**

the SMOTE repetition technique for all minor classes to approach the same number of instances as the major class. The flow chart for the proposed method is shown in Fig. 2.

In Fig. 2, it is seen that the SMOTE process was conducted repeatedly based on the entire minority class in the dataset. The example is in the dermatology dataset. The dermatology dataset consists of 33 features, 366 instances, and six classes. Those six classes are:

1.  Seborrheic dermatitis class that consists of 61 instances.
2.  Psoriasis class that consists of 112 instances.
3.  Lichen planus class that consists of 72 instances.
4.  Chronic dermatitis class that consists of 52 instances.
5.  Pityriasis rosea class that consists of 49 instances.
6.  Pityriasis rubra pilaris class that consists of 20 instances.

The steps of SMOTE proposal are:

1.  Checking the minority class. In the Dermatology dataset, the minority class is the Pityriasis rubra pilaris class, as the total instance is the least compared to others. Therefore, the Pityriasis rubra pilaris class becomes the minority class.
2.  Conducting the SMOTE.
3.  The total instance in the Pityriasis rubra pilaris class doubles the number or becomes 40 instances.
4.  Back to Step 1, if the Pityriasis rubra pilaris class still becomes the minority class, continue to Step 2. If not, the total instance in other classes will be checked to determine which one becomes the next minority class. This should be repeated until all classes experience the SMOTE at least once and the total instance closes to the total instance for the minority class.

## Data preparation

This research was carried out using a computer with an Intel®Core™ i5 processor, 1.6 GHz CPU, 12 GB RAM, and a 64 bit Windows 10 Professional operating system. The development environment was developed using Python, Matlab, and Weka 3.9.2. Meanwhile, eight datasets were used in the UCI Machine Learning Repository (*Dua & Graff, 2019*), including EEG Eye, Cancer (*Dua & Graff, 2019*), Contraceptive Method, Dermatology, Divorce (*Yöntem, Ilhan & Kılıçarslan, 2019*), CNAE-9, Urban Land Cover (*Johnson, 2013*; *Johnson & Xie, 2013*), and Epilepsy (*Andrzejak et al., 2001*). Information and details of each dataset are shown in Table 1.

Each dataset was tested ten times using a random seed with the cross-validation (K-Fold validation 10) process used for the selection of training and test.

## RESULTS

This study conducted feature selection and SMOTE experiments using Weka machine learning tools (version 3.9.2) and MATLAB. The required time and the accuracy performance are divided into two parts: the proposed feature selection and the dataset using the SMOTE process. The performance of the proposed model was compared to other methods such as Correlation Base Feature Selection (CBF) and Information Gain (IG) using a threshold of 0.05 based on the Standard Deviation value (*Prasetiyowati, Maulidevi & Surendro, 2021*) and the original Random Forest (*Breiman, 2001*).

The proposed feature selection technique was the Information Gain (IG) method with a threshold based on the median value, calculated using FFT. The IG transformed with FFT was used to search for the real value. The results of the IG with the threshold were compared with the original Random Forest method. In fact, for IG with a threshold based

**Table 1  Dataset details.**

| Dataset | Number of instance | Number of feature | Number of classess | Dataset status | Area |
|---|---|---|---|---|---|
| EEG eye | 14,980 | 14 | 2 | Imbalance | Life |
| Cancer | 569 | 32 | 2 | Imbalance | Life |
| Contraceptive method | 1,473 | 9 | 3 | Imbalance | Life |
| Dermatology | 366 | 33 | 6 | Imbalance | Life |
| Divorce | 170 | 54 | 2 | Balance | Life |
| CNAE-9 | 1,080 | 857 | 9 | Balance | Business |
| Urban land Cover | 168 | 148 | 9 | Imbalance | Physical |
| Epilepsy | 11,500 | 179 | 5 | Balance | Life |

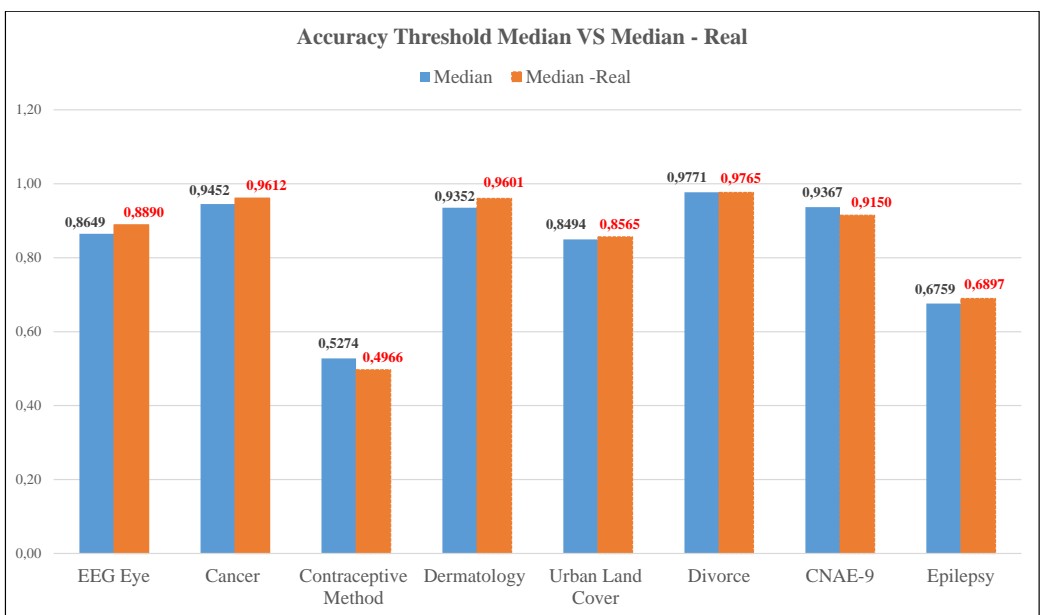

**Figure 3  Comparison of accuracy of median threshold and median-real threshold.**

on the median real (threshold median real), one dataset has a superior accuracy value and another with the same accuracy value. The datasets are the Urban Land Cover and Divorce datasets. If it is compared with the proposal in the previous study (*Prasetiyowati, Maulidevi & Surendro, 2021*), the threshold median real method increases the accuracy in three datasets, namely Cancer, Urban Land Cover, and CNAE-9. In addition, the Divorce dataset has the same accuracy value. However, if the IG threshold median real is compared to the IG threshold median, it is seen that the IG threshold median real results in a better accuracy value. It can be seen in Fig. 3. Five datasets increased. They are EEG Eye, Cancer, Dermatology, Urban Land Cover, and Epilepsy. The threshold value based on the IG

**Table 2  Comparison of accuracy values.**

| Dataset | RF | | CBF | | IG threshold 0.05 | | IG threshold SD | | IG threshold median | | IG threshold median real | |
|---|---|---|---|---|---|---|---|---|---|---|---|---|
| | Accuracy | Num. of feature | Accuracy | Num. of feature | Accuracy | Num. of feature | Accuracy | Num. of feature | Accuracy | Num. of feature | Accuracy | Num. of feature |
| EEG eye | 0.9351 | 14 | 0.7703 | 4 | 0.6316 | 2 | 0.9015 | 10 | 0.8649 | 7 | 0.8890 | 7 |
| Cancer | 0.9633 | 31 | 0.9569 | 12 | 0.9663 | 26 | 0.9439 | 15 | 0.9452 | 16 | 0.9612 | 17 |
| Contraceptive method | 0.5230 | 9 | 0.4874 | 3 | 0.4874 | 3 | 0.5164 | 4 | 0.5274 | 5 | 0.4966 | 5 |
| Dermatology | 0.9701 | 34 | 0.9492 | 15 | 0.9705 | 33 | 0.9743 | 26 | 0.9352 | 17 | 0.9601 | 17 |
| Urban land cover | 0.8536 | 147 | 0.8730 | 28 | 0.8571 | 110 | 0.8476 | 57 | 0.8494 | 74 | 0.8565 | 74 |
| Divorce | 0.9765 | 54 | 0.9653 | 6 | 0.9765 | 54 | 0.9765 | 52 | 0.9771 | 27 | 0.9765 | 27 |
| CNAE-9 | 0.9367 | 856 | 0.8118 | 28 | 0.8756 | 57 | 0.8805 | 65 | 0.9367 | 856 | 0.9150 | 856 |
| Epilepsy | 0.6973 | 178 | 0.6951 | 119 | 0.6973 | 178 | 0.6973 | 178 | 0.6759 | 97 | 0.6897 | 97 |

**Table 3  Comparison of time values.**

| Dataset | Time | | | | | |
|---|---|---|---|---|---|---|
| | RF | CBF | IG Threhold 0.05 | IG Threshold SD | IG Threshold median | IG Threshold median with real |
| EEG Eye | 4.57 | 3.87 | 0.63 | 4.99 | 3.83 | 3.67 |
| Cancer | 0.10 | 0.06 | 0.97 | 0.06 | 0.08 | 0.07 |
| Contraceptive Method | 0.35 | 0.19 | 0.49 | 0.26 | 0.27 | 0.22 |
| Dermatology | 0.07 | 0.04 | 0.05 | 0.04 | 0.05 | 0.05 |
| Urban Land Cover | 0.17 | 0.05 | 0.07 | 0.06 | 0.06 | 0.07 |
| Divorce | 0.02 | 0.01 | 0.02 | 0.02 | 0.01 | 0.02 |
| CNAE-9 | 2.19 | 0.25 | 0.38 | 0.42 | 2.19 | 1.38 |
| Epilepsy | 20.70 | 17.59 | 20.70 | 20.70 | 15.71 | 15.85 |

threshold median real showed an increased accuracy from 0.0071 to 0.0249. The result of the experiment for comparing each method is shown in Table 2.

Figure 3 shows that most datasets produce better accuracy using the median threshold with the transformed IG. Only the Contraceptive Method and Divorce datasets experienced a decrease in inaccuracy. Meanwhile, comparing the aspect of required time, the IG with threshold median real is faster than the RF and IG with threshold Median. The result of the comparison can be seen in Table 3.

Therefore, the method's performance and the Confusion Matrix reference were used to determine each method's Precision, Recall, and F1-Score, as shown in Tables 3 and 4. The displayed Precision, Recall, and F1-Score is a cumulative calculation of 10 seeds given to each dataset. Precision is used to measure the classification accuracy conducted to determine the sensitivity. In comparison F1-Score measures the balance between Precision and Recall.

In the next stage, the researchers conducted the test on the unbalanced dataset. There are five unbalanced datasets: EEG Eye, Cancer, Contraceptive Method, Dermatology, and

**Table 4** Precision, recall and F1- score on random forest, using CBF and IG threshold of 0.05.

| Dataset | Random forest | | | CBF best first | | | IG threshold: 0.05 | | |
|---|---|---|---|---|---|---|---|---|---|
| | Precision | Recall | F1-Score | Precision | Recall | F1-Score | Precision | Recall | F1-Score |
| EEG Eye | 0.9351 | 0.9350 | 0.9353 | 0.7699 | 0.7702 | 0.7700 | 0.6304 | 0.6317 | 0.6310 |
| Cancer | 0.9634 | 0.9632 | 0.9633 | 0.9568 | 0.9568 | 0.9568 | 0.9664 | 0.9664 | 0.9664 |
| Contraceptive method | 0.5192 | 0.5231 | 0.5211 | 0.4873 | 0.4875 | 0.4874 | 0.4873 | 0.4875 | 0.4874 |
| Dermatology | 0.9690 | 0.9691 | 0.9690 | 0.9493 | 0.9492 | 0.9492 | 0.9702 | 0.9704 | 0.9703 |
| Urban land cover | 0.8587 | 0.8534 | 0.8560 | 0.8850 | 0.8809 | 0.8829 | 0.8606 | 0.8571 | 0.8588 |
| Divorce | 0.9780 | 0.9760 | 0.9770 | 0.9656 | 0.9656 | 0.9656 | 0.9780 | 0.9760 | 0.9770 |
| CNAE-9 | 0.9371 | 0.9366 | 0.9368 | 0.7804 | 0.8117 | 0.7852 | 0.8860 | 0.8756 | 0.8808 |
| Epilepsy | 0.6963 | 0.6972 | 0.6967 | 0.6949 | 0.6953 | 0.6951 | 0.6963 | 0.6972 | 0.6967 |

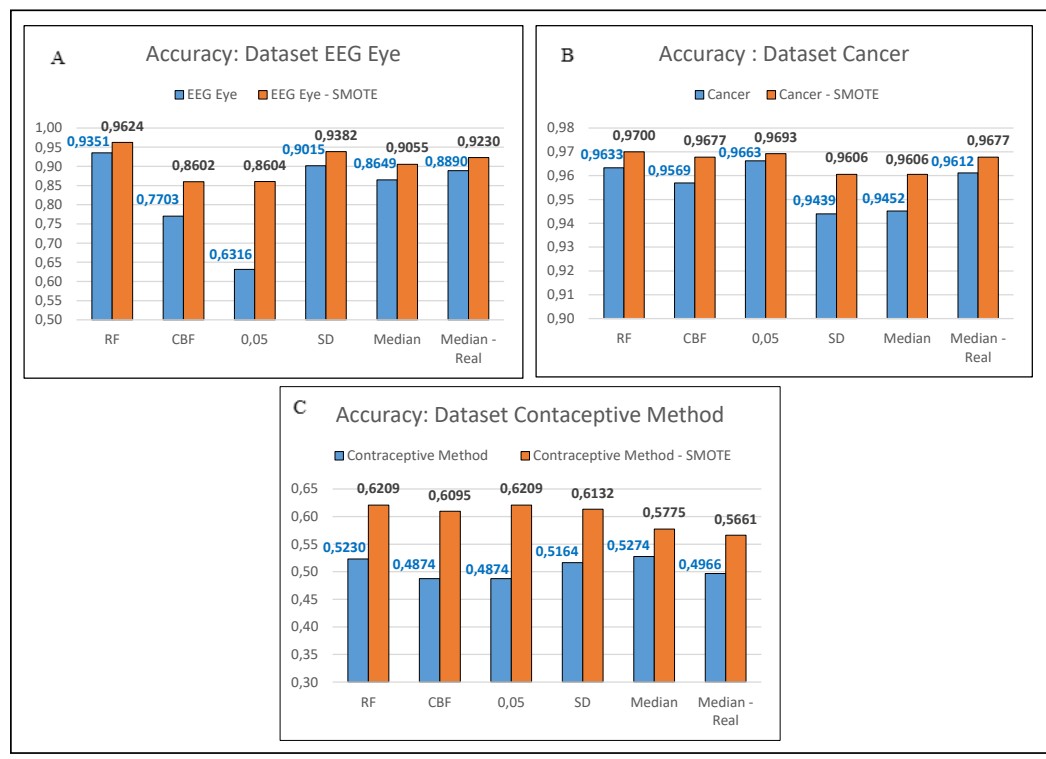

**Figure 4** Comparison of imbalanced and balanced dataset (SMOTE).

Urban Land Cover. Those five datasets were balanced using the SMOTE. The data was suspended on the following datasets: EEG Eye, Cancer, and Contraceptive method, were carried out once. Meanwhile, for the Dermatology and Urban Land Cover datasets, the process of balancing the data was conducted 6 times as the researchers had proposed. The researchers carried this out because there were two minority classes in the dataset, and they needed to be balanced until reaching the major class. Predominantly, the process of balancing the dataset using the SMOTE was conducted repeatedly. Suppose there were more than two minority classes. This process will be conducted repeatedly until all minority

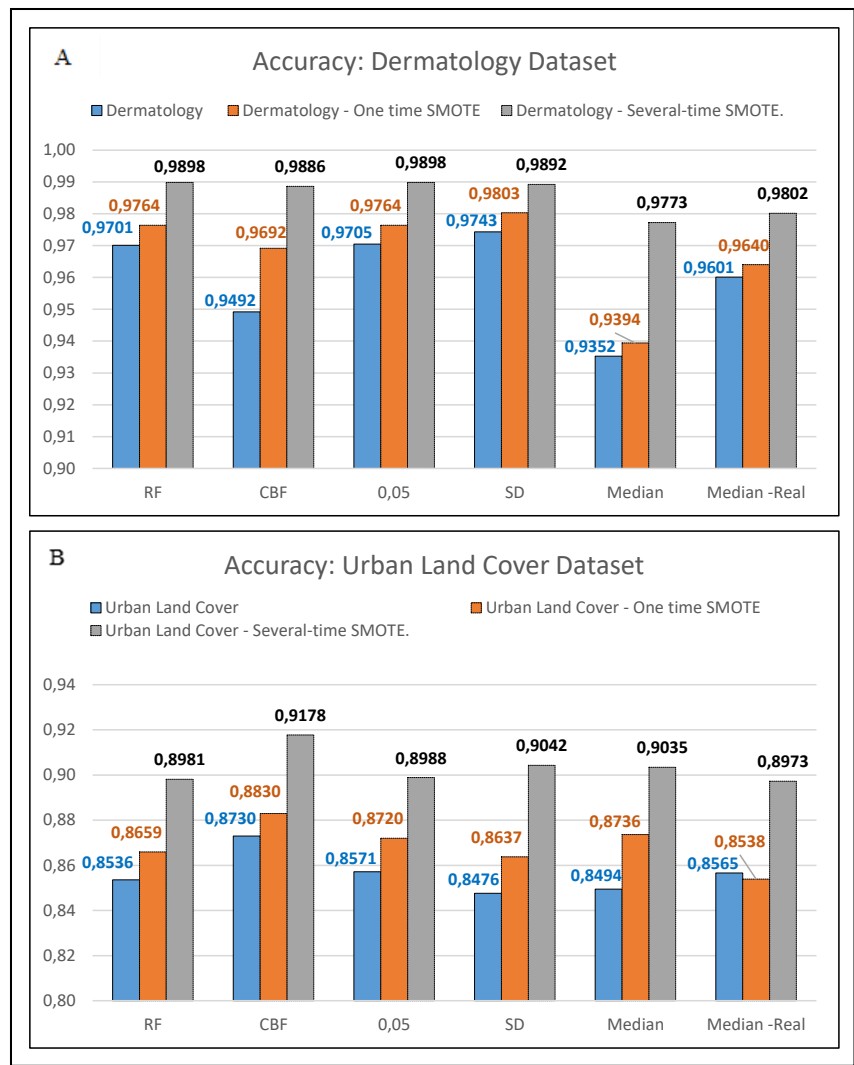

**Figure 5  Comparison between one-time SMOTE and several-time SMOTE.**

classes are close to the major value. The minor value that will be balanced should not be more than the majority class. The results showed that the balanced datasets using SMOTE had better accuracy, as shown in Figs. 4A, 4B and 4C. Similarly, those with two datasets are balanced more than once, as shown in Figs. 5A and 5B.

## DISCUSSION

Based on the eight datasets used here, only the Divorce dataset has the same accuracy value as that resulting from the Random Forest. This accuracy value can be increased by

**Table 5  Precision, recall and F1- score on IG threshold SD, median and median –real.**

| Dataset | IG Threshold: SD | | | IG Threshold: Median | | | IG Threshold: Median - Real | | |
|---|---|---|---|---|---|---|---|---|---|
| | Precision | Recall | F1-Score | Precision | Recall | F1-Score | Precision | Recall | F1-Score |
| EEG Eye | 0.9019 | 0.9013 | 0.9016 | 0.8651 | 0.8650 | 0.8650 | 0.8893 | 0.8891 | 0.8892 |
| Cancer | 0.9437 | 0.9439 | 0.9438 | 0.9450 | 0.9451 | 0.9450 | 0.9611 | 0.9611 | 0.9611 |
| Contraceptive method | 0.5163 | 0.5166 | 0.5164 | 0.5243 | 0.5276 | 0.5259 | 0.4931 | 0.4967 | 0.4949 |
| Dermatology | 0.9743 | 0.9743 | 0.9743 | 0.9389 | 0.9351 | 0.9370 | 0.9600 | 0.9599 | 0.9601 |
| Urban land cover | 0.8530 | 0.8474 | 0.8502 | 0.8537 | 0.8497 | 0.8517 | 0.8614 | 0.8564 | 0.8589 |
| Divorce | 0.9780 | 0.9760 | 0.9770 | 0.9785 | 0.9766 | 0.9775 | 0.9780 | 0.9760 | 0.9770 |
| CNAE-9 | 0.8872 | 0.8806 | 0.8839 | 0.9371 | 0.9366 | 0.9368 | 0.9163 | 0.9152 | 0.9157 |
| Epilepsy | 0.6963 | 0.6972 | 0.6967 | 0.6742 | 0.6759 | 0.6750 | 0.6895 | 0.6898 | 0.6896 |

balancing the dataset using the SMOTE, which is done repeatedly. In this study, SMOTE was repeated several times based on the total majority class in the dataset.

In Figs. 4 and 5, it is seen that the dataset that has been balanced using the SMOTE resulted in a superior accuracy value. In Fig. 5B, the IG method using the threshold Median Real results in a poor accuracy value when conducting one-time SMOTE; however, the accuracy increases when performing multiple-time SMOTE. The researchers conducted the multiple-time SMOTE based on the entire majority class in the dataset. The SMOTE will continue to be conducted as long as the total minority class is below the total majority class. In this study, the multiple-time SMOTE for the Dermatology and Urban Land Cover datasets were conducted six times. The decreased accuracy value in the SMOTE for the Urban Land Cover dataset is because the data generated by the SMOTE did not meet the characteristics of minority classes. Besides, the total instance for each class is not much different.

Besides conducting the SMOTE, the accuracy value can be increased by using the feature that has been transformed using the FFT. This accuracy increase can be seen in Table 2 on the IG threshold Median and IG Threshold Median Real. In the IG threshold median real method, five datasets saw an increase in the accuracy if compared with the IG threshold Median method. EEG Eye, Dermatology, Urban Land Cover, and Epilepsy datasets.

From Table 2 through Table 5, the accuracy value and the F1 score for the datasets, such as the Contraceptive Method and the Epilepsy datasets, decrease. The factor is that the total feature used here is less. In the Contraceptive method, the accuracy decreased since the entire feature used here was five out of nine existing features. The Epilepsy dataset also used 97 features out of 178 available features. Meanwhile, all datasets available in Tables 4 and 5 are the datasets that have not been processed using the SMOTE. The SMOTE is not required to be conducted in three datasets, Divorce, CNAE-9, and Epilepsy, as those three datasets are balanced already.

Even though the aspect of accuracy decreases, the part of required time for the IG threshold, median real method needs more diminutive than the Random Forest without

feature selection. The time difference between feature selection with the IG threshold median real and the original Random Forest is between 0.03 and 4.85 s.

## CONCLUSIONS AND FUTURE WORK

Based on the testing, it can conclude that the Information Gain (IG) with a threshold median three times superior to the accuracy generated by the Random Forest, especially in the data aggregate of Contraceptive Method, Divorce, and CNAE-9. Nevertheless, the accuracy value for the IG with threshold median real is higher than the threshold accuracy value based on the Median score. Five datasets have an accuracy value higher than that of the IG Threshold Median; those include EEG Eye, Cancer, Dermatology, Urban Land Cover, and Epilepsy. The increase in this accuracy value applies to both the original dataset and the dataset that has been balanced using the SMOTE. It can be inferred that FFT and SMOTE can increase the accuracy value, mainly if the SMOTE is conducted repeatedly according to what has been proposed by the researchers.

Even though the accuracy value in the feature selection with IG threshold median real is less superior to that of the original Random Forest, this method is superior in speed. The time required in this method is less than that of the original Random Forest.

The subsequent study that needs to be considered is using the two-level feature selection based on the roadmap that the researcher suggests in Fig. 1. The next study that needs to be considered is using multilevel feature selection based on the roadmap the researcher guides in Fig. 1. In addition, selecting more informative features also needs to be considered.

## ACKNOWLEDGEMENTS

We would like to thank the Institut Teknologi Bandung and Universitas Multimedia Nusantara (UMN) for providing the time and access to resources needed to conduct this research.

### Funding

The authors received no funding for this work.

### Competing Interests

The authors declare there are no competing interests.

### Author Contributions

- Maria Irmina Prasetiyowati conceived and designed the experiments, performed the experiments, analyzed the data, performed the computation work, prepared figures and/or tables, authored or reviewed drafts of the article, we tried a new method to determine the threshold on Information Gain. The cut-off (threshold) obtained is the median value of the conversion of the IG value into the FFT value, and approved the final draft.

- Nur Ulfa Maulidevi analyzed the data, performed the computation work, prepared figures and/or tables, authored or reviewed drafts of the article, and approved the final draft.
- Kridanto Surendro analyzed the data, performed the computation work, prepared figures and/or tables, authored or reviewed drafts of the article, and approved the final draft.

## Data Availability

The data and code are available in the Supplemental Files.

## Supplemental Information

Supplemental information for this article can be found online at http://dx.doi.org/10.7717/peerj-cs.1041#supplemental-information.

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
