# Peer review of "The accuracy of Random Forest performance can be improved by conducting a feature selection with a balancing strategy"

_PeerJ Computer Science, doi:10.7717/peerj-cs.1041_

## Round 0.1 · original submission · Major Revisions

· Academic Editor

Major Revisions

A major revision is needed before further processing the paper. Please provide a detailed response letter. Also please note that I do not expect you to cite any recommended references unless it is essential.

Reviewer 1 ·

Basic reporting

no comment

Experimental design

no comment

Validity of the findings

no comment

Additional comments

1. According to the Abstract and Introduction sections, the purpose of this paper is to improve the accuracy and processing speed of random forest, but the actual research only reflects the accuracy and does not analyze the processing speed improvement.
2. The parameters or hyperparameters of machine learning algorithm also have a great impact on the accuracy of the model. This work mainly studies the influence and improvement methods of feature screening and unbalanced data on the model accuracy of random forest, but does not study the influence of algorithm parameter or hyperparameter optimization methods (such as Bayesian hyperparameter optimization, refer to sun et al. 2020, 2021, Doi: 10.1016/j.geomorph.2020.107201, Doi: 10.1016/j.enggeo.2020.105972), and the present work cannot cover the subject. It is suggested to add "feature selection" and "SMOTE" to modify the title.
3. Does the features selection methods proposed in this work have comparative advantages with the widely used recursive feature elimination (RFE) methods, such as Zhou et al. (2021) used in DOI: 10.1016 / j.gsf. 2021.101211?
4. For the Conceptual method and Epilepsy data sets, the model effect is not good with lower accuracy and F1 score, whether it is improved or not. The possible causes should be analyzed in the discussion section.
5. There are meaningless to use broken lines in figure 2-4. It is suggested to change broken lines to histograms.
6. The number of factors of each data set by using each feature selection method are recommended to be expressed in a table.
7. Compared with table 2, the expression of conclusion a and b is inaccurate and confusing, which is recommended to be modified.
8. Table cs-68916-7_CNAE in the Supplemental Files seems incomplete, missing feature X1.
9. It is strongly recommended to add the content of the Discussion section.
10. See the Attachment PDF File for other writing errors.

Annotated reviews are not available for download in order to protect the identity of reviewers who chose to remain anonymous.

Reviewer 2 ·

Basic reporting

This paper proposes a method to improve accuracy performance in random forest. Some important comments:
1. The main contribution of this paper is not explained, for example the difference with previous studies.
2. Explanation of the flow of thought on how IG uses the standard deviation (SD) as a threshold, because the standard deviation is a measure of the spread of data, not the concentration of data. Is the purpose of using SD as a threshold described in the paper the same as the research of Sindhu and Radha, 2020?
3. In line 66, it is explained that the IG threshold using the standard deviation is less successful in determining the threshold when considering data balance, then in this study using the median. How to explain the flow of thought?
4. It is necessary to explain the flow of thought why IG needs to be transformed using FFT, then to determine the threshold only use the real part. What does the real and imaginary part after the FFT process represent?

Experimental design

1. In this study using several unbalanced datasets, for performance measurement why only use accuracy, it is better to add others.
2. Why for data separation using k-fold cross validation, not using a certain proportion between training and testing?
3. In Figure 1, a flow chart that explains the SMOTE process for those with more than two classes, needs to be described in more detail, preferably with an example.

Validity of the findings

1. In Figure 4, the term first-SMOTE appears, what does it mean, where is it explained?
2. In the results and discussion, the results obtained need to be explained more comprehensively based on the results of previous studies. For example in Figure 4A, why does the dermatology-first smote accuracy decrease sharply using the median threshold.

Additional comments

1. Split data using K-Fold validation instead of K-Folt validation (typo)
2. It should be added in the background, the difference between the current research and previous research conducted by the researcher (it can be described as a kind of research roadmap).
3. In the conclusion section, write a comprehensive conclusion, not just in the form of points without the connection between points.

---

## Round 0.2 · Minor Revisions

· Academic Editor

Minor Revisions

Reviewer 1 still has some comments. Please address them and resubmit the paper with a response letter. Thanks.

Reviewer 1 ·

Basic reporting

1) I think the 3 references mentioned in Concern #2 and # 3 of the last review comments should be added to the references to elaborate the research background in the Introduction section.

Experimental design

None

Validity of the findings

2) Due to feature selection and SMOTE are common model optimization method in machine learning, it suggested to further strengthen the major contribution of this work concern on the influence and improvement methods of feature selection and SMOTE on the model accuracy and processing speed of random forest in the Introduction and Discussion section.

Additional comments

None

Reviewer 2 ·

Basic reporting

All previous concerns have been addressed.

Experimental design

The corrections have been done as per requirement.

Validity of the findings

All suggestions have been fulfilled

Additional comments

The authors must improve the grammar of English writing

---

## Round 0.3 · accepted · Accept

· Academic Editor

Accept

The paper can be accepted. Congratulations.

Reviewer 1 ·

Basic reporting

All previous concerns have been addressed.

Experimental design

None

Validity of the findings

All previous concerns have been addressed.

Additional comments

None